# The Hallmarks of Cervical Cancer: Molecular Mechanisms Induced by Human Papillomavirus

**DOI:** 10.3390/biology13020077

**Published:** 2024-01-27

**Authors:** Pedro Rosendo-Chalma, Verónica Antonio-Véjar, Jonnathan Gerardo Ortiz Tejedor, Jose Ortiz Segarra, Bernardo Vega Crespo, Gabriele Davide Bigoni-Ordóñez

**Affiliations:** 1Laboratorio de Virus y Cáncer, Unidad de Investigación Biomédica en Cáncer of Instituto de Investigaciones Biomédicas, Universidad Nacional Autónoma de México (IIB-UNAM), Mexico City 14080, Mexico; prosendo.chalma@gmail.com; 2Unidad Académica de Posgrado, Universidad Católica de Cuenca, Cuenca 010101, Ecuador; jonnathan.ortiz@ucacue.edu.ec; 3Laboratorio de Biomedicina Molecular, Facultad de Ciencias Químico Biológicas, Universidad Autónoma de Guerrero, Chilpancingo 39090, Guerrero, Mexico; 11335@uagro.mx; 4Carrera de Bioquímica y Farmacia, Universidad Católica de Cuenca, Cuenca 010101, Ecuador; 5Carrera de Medicina, Facultad de Ciencias Médicas, Universidad de Cuenca, Cuenca 010107, Ecuador; jose.ortiz@ucuenca.edu.ec (J.O.S.); bernardo.vegac@ucuenca.edu.ec (B.V.C.); 6Carrera de Laboratorio Clínico, Facultad de Ciencias Médicas, Universidad de Cuenca, Cuenca 010107, Ecuador

**Keywords:** HPV, uterine cervical cancer, viral load, viral physical state, integration, methylation, metastasis

## Abstract

**Simple Summary:**

The role of human papillomavirus (HPV) in cervical carcinogenesis is widely documented; however, with an increasing number of scientific publications on the molecular and cellular mechanisms activated by the virus and, specifically, by high-risk HPVs (HR-HPVs) that are involved in the development of uterine cervical cancer (CaCU) and its precursor lesions, we consider it is important to present a review of scientific articles that address ten of the mechanisms associated with at least seven of the fourteen hallmarks of cancer recently proposed. Understanding the mechanisms activated by HR-HPVs in the context of the distinctive physiological capabilities of cancer will allow the identification of clinically relevant biomarkers to improve the diagnosis and treatment of CaCU.

**Abstract:**

Human papillomaviruses (HPVs) and, specifically, high-risk HPVs (HR-HPVs) are identified as necessary factors in the development of cancer of the lower genital tract, with CaCU standing out as the most prevalent tumor. This review summarizes ten mechanisms activated by HR-HPVs during cervical carcinogenesis, which are broadly associated with at least seven of the fourteen distinctive physiological capacities of cancer in the newly established model by Hanahan in 2022. These mechanisms involve infection by human papillomavirus, cellular tropism, genetic predisposition to uterine cervical cancer (CaCU), viral load, viral physical state, regulation of epigenetic mechanisms, loss of function of the E2 protein, deregulated expression of E6/E7 oncogenes, regulation of host cell protein function, and acquisition of the mesenchymal phenotype.

## 1. Introduction

According to data published by the International Agency for Research on Cancer of the World Health Organization (IARC-WHO; Globocan 2020), worldwide, uterine cervical cancer (CaCU) is the fourth most common cancer and the third cause of death in women. In Latin America, it is not only the third most common cancer, but also the third cause of death in the female population [1,2,3].

Seventeen years have passed since the first marketing of vaccines against the human papillomavirus (HPV) was authorized. However, a recent report from the WHO indicates that, to date, only 60% of the member states of the organization have introduced the HPV vaccine in their national vaccination schedule and that until 2021, only 13% of girls in the world had completed the planned vaccination schedule [4]. Therefore, CaCU continues to be a global public health problem, with a particularly high burden in low- and middle-income countries (LMICs), such as Mexico and Ecuador, where the incidence and mortality rate of CaCU occupy an alarming second place [1].

For this reason, in the current review, we present an outline of a series of distinctive molecular multi-step mechanisms that are involved in the carcinogenic process of the cervix and that could be considered as molecular targets for the timely treatment of neoplasms caused by HPV.

## 2. Human Papillomavirus Infection

HPVs are small icosahedral viruses, approximately 50 to 60 nm in diameter, non-enveloped, containing a circular double-stranded DNA genome (between 7000 and 8000 bp) (see Figure 1), infecting mucosal and skin epithelia in a specific manner and inducing cell proliferation [5,6,7].

The HPV genome is organized similarly to chromatin [17] and is divided into three functional regions. The first is a “non-coding upstream regulatory region”, also known as the long control region (LCR) or upper regulatory region (URR). This region contains the p97 core promoter along with cis-enhancer elements that include binding sites for the viral proteins E1 (E1BS) and E2 (E2BS)—required for the commencement of HPV replication—and binding sites for several cellular transcription factors, including Sp1, YY1, TEF-10, AP1, Oct-1, NF1, KRF-1 and glucocorticoid response elements (GREs), required for the initiation of transcription [8,9,10,11,12,13,14,15,18]. The second is called the “early (E) region” and consists of the open reading frames (ORFs) for E1, E2, E4, E5, E6 and E7, where the E1, E2 and E4 proteins are mainly associated with replication, transcription, and viral integration. The E5 protein regulates cell proliferation and apoptosis and facilitates the activity of E6 and E7, while E6 and E7 act as oncoproteins and are associated with cancer development and progression [19,20,21]. The third region is known as the “late region (L)”, comprises 40% of the viral genome and includes the ORFs L1 and L2 that encode the viral capsid proteins [22].

In 1983, Harald Zur Hausen and his working group established, for the first time, the relationship between HPV and CaCU [23], but it was not until 1995 that the IARC-WHO evaluated and considered HPV as a biological agent with carcinogenic risk for humans [24,25]. Currently, there are 229 different types of HPV [26,27], classified by the IARC-WHO into three groups according to their oncogenic potential. Group 1, referred to as ‘carcinogenic or oncogenic’ (also called high-risk or HR-HPV), includes types 16, 18, 31, 33, 35, 39, 45, 51, 52, 56, 58 and 59. Of these types, HPV 16 and 18 are considered to be the most important for their association with CaCU. Group 2, referred to as intermediate-risk, is subdivided into Group 2A, called ‘probably carcinogenic’, which includes only HPV 68, and Group 2B called ‘possibly carcinogenic’, which includes types 26, 53, 66, 67, 69, 70, 73 and 82. Group 3, called ‘not classifiable as carcinogenic (low risk)’, includes types 6, 11, 40, 42, 53, 54 and 57 [28,29,30,31,32].

To date, only viral types in Group 1 (HPVs or HR-HPVs) have been associated with the development of both CaCU and other types of cancer, including anogenital cancers (penis, vulva, and vagina) and cancers of the head and neck [28,33,34,35]. In addition to this, a recent study showed that almost one in three men worldwide are infected with at least one type of genital HPV and around one in five men are infected with one or more types of HR-HPV (14). This indicates that men frequently harbor genital HPV infections, emphasizing the importance of incorporating men in efforts to control HPV infection and reduce the incidence of HPV-related diseases in both men and women [36].

## 3. Cellular Tropism

The uterine cervix is divided into three regions, i.e., the exocervix (also called ectocervix), the endocervix and the squamocolumnar junction (SCJ) or transformation zone (considered a misnomer for a benign process, since the term “transformation” is currently used in oncology to refer to malignant neoplastic transformation). The ectocervix is composed of a non-keratinized stratified squamous epithelium and contains four phenotypically distinct cell populations: epithelial stem or stem-like cells, located in the basal and parabasal layer, and differentiated cells, located in the intermediate and superficial layers. The endocervix is lined by a single layer of mucinous columnar cells (also referred to as columnar epithelium or glandular epithelium). The SCJ is the transition area between the ectocervix and the endocervix and consists of endocervical squamous metaplasia cells, which include endocervical reserve cells (a specialized type of tissue stem cell) and possibly cuboidal cells located, more precisely, in the squamocolumnar junction, which have the capacity to divide and renew [37,38,39,40,41].

John Doorbar [42] extensively described the cellular tropism of HPV. His findings allowed us to establish that in the case of non-keratinized stratified squamous epithelium, the presence of a micro-wound is required that allows infectious virions to access the basal layer and specifically infect stem-like cells. Once infected, the stem-like cells form a reservoir of infection, and in these cells, the viral genome is maintained in an episomal state with a low copy number; as the cells divide, they produce daughter cells that are pushed towards the epithelial surface, giving rise to transient productive infections possibly progressing to high-grade neoplasia or squamous cell carcinoma [42,43,44]. Conversely, it is suggested that HPV can also infect mucinous columnar cells, reserve cells and cuboidal cells located in the SCJ and endocervix, where infection of these cell types is associated with different patterns of disease progression and the development of adenocarcinoma [45,46,47,48].

Importantly, most research models of HPV-associated cervical carcinogenesis focus on the non-keratinized stratified squamous epithelium, while the columnar epithelium of the endocervix and the metaplastic epithelium (which contains reserve cells and cuboidal cells) of the SCJ have received less attention. Evidence of this difference in research focus is that the mechanism by which HR-HPV infects stem-like cells is widely known. Specifically, these cells are characterized by expressing α6β4 integrin receptors, the epidermal growth factor receptor (EGFR), the keratinocyte growth factor receptor (KGFR) and heterotetrameric annexin A2/S100A10 (A2t) receptors, which are necessary for the entry of virions into the cell [49,50,51]. On the other hand, in the case of the epithelia of the endocervical region and the SCJ, it is only known that reserve cells that have a CK17/p63 phenotype are easily accessible targets for HPV infection [52,53,54,55].

## 4. Genetic Predisposition to Cervical Cancer

Several genome-wide association studies (GWASs) in different populations have provided evidence that there is a certain genetic susceptibility associated with the development of CaCU. A GWAS study of the British population identified certain single-nucleotide polymorphisms (SNPs) in the PAX8, CLPTM1L and HLA genes, with the SNPs rs10175462 in PAX8, rs27069 in CLPTM1L and rs9272050 in HLA-DQA1 being strongly associated with the risk of developing CaCU [56]. Another GWAS study of the Saudi population determined that the SNPs T10C in the GFB1 gene and G399A in the XRCC1 gene were associated with a 1.5-fold increase in the risk of developing CaCU [57]. Some studies reported that functional SNPs in codon 72 of TP53 and SNP609 in the NQO1 gene are associated with the risk of developing CaCU [58]. Finally, the homozygous CC genotype in the SNP rs4646903 of the CYP1A1 gene—which participates in genetic repair mechanisms—and the CT heterozygous genotype in the SNP rs1801133 of the MTHFR gene—which participates in cellular detoxification—are not only associated with the development of CaCU and high-grade dysplasia, but may also contribute to disease progression [59].

## 5. Viral Load

Initially, the detection of HR-HPV viral load was used as an additional test to relate the viral copy number to an active infectious process and reduce false-negative results in HPV diagnostic assays [60]. Other studies correlated the viral load of HR-HPV with the age of the patient, histological severity, multiple viral types, the area of the cervical lesion and the sampling method (endocervical and exocervical) [61,62]. The viral load has also been proposed as a significant marker of progression towards precancerous lesions, that is, as the viral load increases, the risk of cervical lesions increases. The risk is further enhanced if the HPV genotype is high-risk, if the viral infection is persistent during the cervical disease, and if recurrent infections are contracted with different HPV genotypes [63,64,65,66]. For example, recent studies reported that a high viral load of HR-HPVs, specifically HPV16, is significantly related to a higher risk of developing CIN2+, suggesting that viral load could be a relevant biomarker to identify women with high susceptibility to developing precancerous lesions in the uterine cervix [67,68,69].

## 6. Viral Physical State

At the outset of viral infection in the stem-like cells of the ectocervix, the HPV genome persists as a naked nucleic acid (also called an episome), and it depends on the host cell to enable replication. This occurs in the nucleus, with the genome replicating as an extrachromosomal element each time the cell divides [70]. As infected cells differentiate and move towards the surface of the epithelium, high levels of viral DNA are replicated, packaged into virions, and released from the surface of the epithelium as virus-laden squamous cells [47], thus completing the viral life cycle.

Conversely, it was reported that during the infectious phase, HPV can remain in its episomal form, integrate into the genome of the host cell or even be present in a coexisting state (episomal/integrated) [71,72,73,74]. In 1987, Awady and collaborators analyzed the integration of HPV16 in the SiHa cell line and reported, for the first time, the deletion of 251 nucleotides of the viral sequence within the ORFs E2 and E4 (viral integration site), and a deletion in chromosome 13 of 4.8 kb of the cellular genomic sequence (integration site in the cellular genome) [75]. Another study on invasive CaCU samples carried out by Kalantari and collaborators reported that the HPV16 genome was integrated between the E1 and E2 regions and that the integration site in the cellular genome was located in the chromosomal regions 1q25, 3q28, 6p25, 11p13 and 18q22 [76]. In Figure 1 illustrates the region of breakage and integration in the HPV16 genome.

By utilizing the Capture-HPV NGS method using tumor biopsies of patients with CaCU, it was determined that HR-HPVs are inserted into intact and repeated regions of the cellular genome, specifically in MYC, NUDT15, MED4, ITM2B, RB1 loci, LPAR6, KLF5, KLF12, PIBF1, RB1, AKT3, SST, ID1, LPP, AFF3, BCL6, CCAT1, CCAT2, RAB11A, RAB22A, MAST4 and MAP2, among others [77]. By considering these findings coupled with results from women with normal cytology, those positive for HR-HPV and those with an integrated viral physical status [78,79], we can hypothesize that the integration of the viral genome at the start of infection in the target cell, will not only lead to genomic instability but also induce the development of a malignant neoplastic process (see Figure 2).

## 7. Initiation of Epigenetic Mechanisms

In general, it is understood that tumor-associated DNA viruses are organized into nucleosomes to regulate the expression of their genes through histone modifications, particularly, histone acetylation. Methylation of the viral genome occurs during infection as a cellular defense mechanism against the entry of foreign genomes [80,81]. In this section, we highlight the most relevant epigenetic mechanisms induced by HPV during carcinogenesis.

### 7.1. Activation of the Cellular Methylation Machinery

The first studies referring to HPV DNA methylation were carried out in the LCR of the viral genome using different techniques. One of these involved methylation-specific PCR (MSP), which made it possible to report different methylation states (hypomethylated, hemimethylated and hypermethylated) depending on the amplification specificity of the primers [82,83,84,85,86,87]. Another comprised bisulfite sequencing PCR (BSP), which allowed reporting of the methylation patterns or methylation frequencies (%) of each of the CpG sites of the LCR [88,89,90,91,92,93,94].

Based on different publications alluding to the methylation of the LCR of HR-HPVs and given the premise that the methylation machinery is activated as a defense mechanism against foreign genomes, the question is raised as to how HR-HPVs activate the cellular methylation machinery. Based on previous studies, it can be inferred that HPV activates the methylation machinery through two physical mechanisms. The first occurs during HPV infection, when the entry of the viral particles into the target cell activates the methylation of the viral DNA via DNA methyltransferase 1 (Dnmt1). Following the differentiation of the host cell, the viral LCR is hypomethylated to regulate the expression of viral genes during the normal viral life cycle in the non-keratinized stratified squamous epithelium [95,96,97,98]. The second mechanism involves the integration of the viral genome, which activates the cellular methylation machinery again. However, during this process, methylation occurs only in regions where the viral genome is integrated into tandems, and the distal viral genomes are transcriptionally active and hypomethylated [94,99,100,101].

Findings published by Fernández et al. [102] on the DNA methylomes of HR-HPVs suggest that the viral load and the integration of the viral genome could play an important role in inducing different methylation patterns as the disease evolves. For example, in this study, the HeLa (derived from an adenocarcinoma that contains between 10 and 50 integrated copies of HPV18 per cell), SiHa (derived from a grade II cervical squamous cell carcinoma containing from 1 to 2 integrated copies of HPV16 per cell), and Ca Ski (derived from a cervical squamous cell carcinoma that contains between 500 and 600 integrated copies of HPV16 per cell) cell lines were used [102,103,104], and it was found that the HPV18 genome in HeLa cells is mostly demethylated, with site-specific methylation only in the E2 and L1 regions, while in SiHa cells, the HPV16 genome is demethylated in the LCR, E6, E7 and E1 regions, with methylation in the E2/E4, E5, L2 and L1 regions. Interestingly, in Ca Ski cells, it was found that the majority of HPV16 genomes are hypermethylated, and only a few are hypomethylated, suggesting that the latter are those which are transcriptionally active [102].

### 7.2. Histone Rearrangement

Favre and collaborators were the first to describe that the HPV genome is associated with the canonical histones H2A, H2B, H3 and H4 [17]. It was reported that the E2, E6 and E7 proteins of HR-HPV have the capacity not only to bind to the CBP/p300 coactivator complex and inhibit its histone acetyltransferase (HAT) activity, but also to block the ability of p300 to activate p53-responsive promoter elements. This results in the deregulation of cellular signaling, which decreases genome stability and favors the cellular transformation process [105,106,107,108].

## 8. Loss of E2 Protein Function

It is generally understood that the oncogenic HPV E2 protein is a negative regulator of the expression of the E6 and E7 oncogenes [72]. It is understood that the loss of E2 function can occur in two ways: the first is through the integration process, where breaks in the E1/E2 regions lead to the functional loss of the E2 gene [72,76,109,110]; and the second involves the methylation of the CpG sites located in the E2BSs of the HPV LCR, specifically, E2BS1, E2BS3 and E2BS4, which results in the activation of the p97 promoter and the subsequent loss of the repressive function of the E2 protein on the transcription of E7/E6 [96,111,112,113,114]. Therefore, the loss of E2 function could be considered a key step in carcinogenesis.

## 9. Deregulated Expression of the E6/E7 Oncogenes

Various studies reported that the loss of E2 function—either due to the phenomenon of viral genome integration or due to the methylation of the E2BSs in the HPV LCR—is associated with the overexpression or the aberrant expression of E6 and E7 [102,115,116,117]. However, these same studies mentioned that there was no significant difference when comparing the expression levels of the E6/E7 oncogenes in samples that expressed E2 and contained HPV genomes in a purely episomal state or in a coexisting state, with those in samples that contained HPV genomes in a purely integrated state, without E2 expression. This indicates that methylation at specific sites of the E2BSs in the LCR plays an important role not only in the loss of E2 function in those samples harboring transcriptionally active E2 genes, but also in regulating the expression level of E6/E7. This suggests that the overexpression of the E6/E7 oncogenes can be favored only in cases where the following criteria are met: (1) there is a high number of viral genomes in the episomal state with intact E2 genes and with site-specific methylation in E2BS-I and -II; (2) there is a low or moderate viral load, and the viral genomes are integrated at distal sites in a single copy and probably under the control of strong promoter regions in the host cell genome [102,117].

## 10. Regulation of Host Cell Protein Function

It was reported that the E6 oncoprotein of HR-HPV can evade cell death by apoptosis through two pathways. The first is through the proteasomal degradation of p53 via its association with the ubiquitin ligase UBE3a (E6AP) [118,119,120]. The second is through the interaction of E6 with hADA3—a protein that functions as a coactivator of p53-mediated transactivation for a variety of target promoters—where E6 induces the degradation of hADA3, thus inactivating the function of p53 and overriding the arrest of p14ARF-induced cell growth, despite the presence of normal levels of p53 [121,122]. Conversely, it is widely accepted that the E7 oncoprotein of HR-HPVs plays two main roles to induce the transforming and proliferative process in cells. Firstly, it binds with members of the retinoblastoma protein (pRb) family, such as p107 and p130 [123], which promotes the transcriptional activity of E2F transcription factors, thus regulating cell cycle entry and the progression from the G1 phase to the S phase of the cell cycle [124]. Secondly, it destabilizes pRb via degradation through the ubiquitin–proteasome pathway, leading to oncogenic transformation [125].

Table 1 exemplifies host cell proteins that interact with HPV proteins and their effect on different cellular mechanisms.

## 11. Acquisition of the Mesenchymal Phenotype

An established feature of solid tumors which are not associated with oncogenic viruses is the acquisition of a mesenchymal phenotype. This is characterized by the overexpression of N-cadherin, vimentin, fibronectin, Twist, FOX C2, SOX 10, MMP-2, MMP-3, MMP-9, Snail and Slug (currently designated as Snai1 and Snai2, respectively, by the HUGO Gene Nomenclature Committee) and a decrease in the expression of E-cadherin (currently designated as CDH1 by the HUGO Gene Nomenclature Committee), desmoplakin, cytokeratin and occludin [203,204]. Nevertheless, Hellner and collaborators [205] reported that both E6 and E7 induced the expression of N-cadherin and that the expression of E7 in primary human foreskin keratinocytes (HFK) induced elevated levels of vimentin and fibronectin, as well as reduced levels of CDH1, while the levels of the regulators Twist and Snai1 remained unchanged. Another study performed in a NIKS cell model demonstrated that HPV E7 not only induced the expression of Dnmt1 but also was associated with the suppression of CDH1 expression. However, despite the expression of Dnmt1, no methylation of the CDH1 promoter region was observed, nor was any alteration observed in the expression of the negative regulators of CDH1 (Snai1/Snai2) [206]. Furthermore, a study utilizing Madin–Darby canine kidney (MDCK) cells proposed that both E6 and E7 of HPV16 could play an important role in the epithelial–mesenchymal transition (EMT) process by inducing the expression of the transcriptional factors Snai2, Twist, ZEB1 and ZEB2 and reducing CDH1 expression [207].

It is widely accepted that a common characteristic of tumors associated with oncogenic viruses—such as Epstein–Barr virus (EBV), human papillomaviruses (HPV) and hepatitis B and C viruses (HBV, HCV)—when acquiring the mesenchymal phenotype, is the suppression of CDH1 expression [208]. The interaction of viral oncoproteins with Dnmt1 plays an important role in suppressing the expression of CDH1 via methylation of its promoter region [209]. However, since reports showed that HPV may or may not methylate the CDH1 promoter region—despite inducing Dnmt1 overexpression and promoting its activity [206,210,211,212]—and given the fact that HPV does not significantly alter the expression of negative regulators of CDH1 [203,205,206], the question arises as to how HPV participates in regulating CDH1 expression and in inducing a mesenchymal phenotype.

Following the premise that both E6 and E7 of HPV have the ability to induce the expression of, bind to and stimulate the methyltransferase activity of Dnmt1 [185,189] and that E7 interacts with Mi2β, as well as with HDAC1 and HDAC2, to modulate the expression of cellular genes and viral genes by chromatin rearrangement [157,213], and knowing that the cell lines HeLa (adenocarcinoma of the cervix), SiHa (squamous cell carcinoma) and Ca Ski (squamous cell carcinoma of the cervix) are representative of the most common types of CaCU with HR-HPV infection and have different viral load and different epithelial origin, our working group previously reported that HR-HPVs can induce a mesenchymal phenotype by negatively regulating the expression of CDH1 through different pathways in which E7, Snai1 and epigenetic mechanisms are involved [214]. For example, in HeLa cells, it was found that E7 suppresses the expression of CDH1 via total methylation of the CDH1 promoter region and overexpression of Snai1, most likely forming an E7/Snai1/Dnmt1 repressive complex. Similarly, in SiHa cells, methylation of 17.65% of the CDH1 promoter region was observed, with significant expression of Snai1, which gave rise to a slight expression of CDH1; this suggests that CDH1 may be regulated by a complex consisting of E7/Snai1/HDAC1. Conversely, the Ca Ski cell line did not exhibit a mesenchymal phenotype, since it showed a high level of expression of CDH1, without methylation of its promoter region, and low levels of expression of Snai1 and Snai2 [214]. Our results demonstrated that HR-HPVs can regulate the expression of TEM markers in different ways, most likely depending on the infected epithelium and the viral load.

Figure 3 provides a representative diagram of the main molecular hallmarks that have been reported during the carcinogenic process of CaCU.

## 12. Conclusions

HR-HPVs, through their oncoproteins E6 and E7, are responsible for the cellular changes linked to the development of CaCU. During the process of viral genome replication—whether as an episome, integrated or coexisting in the episomal and integrated states—modifications are generated in the host cell machinery, which induce genomic instability and the development of the carcinogenic process. In this review, we described ten mechanisms activated by HR-HPVs during cervical carcinogenesis, which are broadly associated with at least seven of the fourteen distinctive physiological capacities of cancer in a newly established model [215]. Specifically, the mechanisms involved are among those that promote epigenetic modifications, instability in the host cell genome, sustained proliferative signaling, replicative immortality, resistance to cell death, evasion of the immune response and activation of invasion and metastasis. Therefore, improved understanding of the viral oncogenic mechanisms will allow us to develop new tools for the early diagnosis of cervical lesions and to identify other therapeutic targets with a focus on the early phases of cervical malignancy.

## Figures and Tables

**Figure 1 biology-13-00077-f001:**
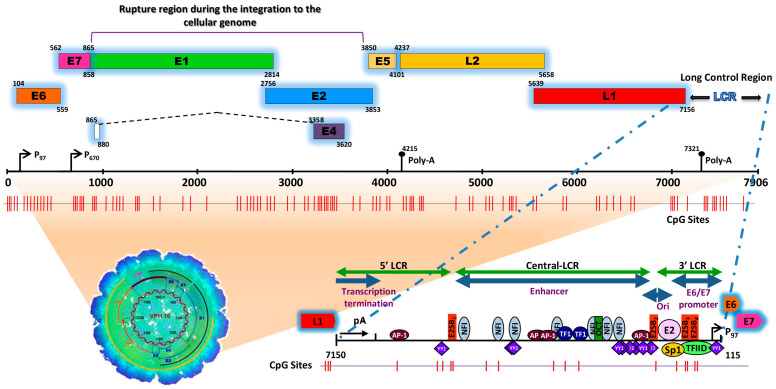
Schematic representation of the structure of the HPV type 16 (HPV16) genome and its long control region (LCR) as a representative model of genital HPVs. The red vertical lines indicate the position of the 112 CpG sites along the viral genome. The bottom of the schematic illustrates the segments into which the LCR is divided as well as the cellular transcription factors that bind to it [8,9,10,11,12,13,14,15]. To illustrate the genomic structure of HPV16, the latest update of the genomic sequence was used, with NCBI Reference Sequence NC_001526, as well as PISMA software for the localization of each of the CpGs sites [16] and Vector NTI^®^ Express Designer Software v1.5.1 (Thermo Fisher Scientific Inc., Waltham, MA, USA) for the identification of the ORFs of each of the HPV16 genes.

**Figure 2 biology-13-00077-f002:**
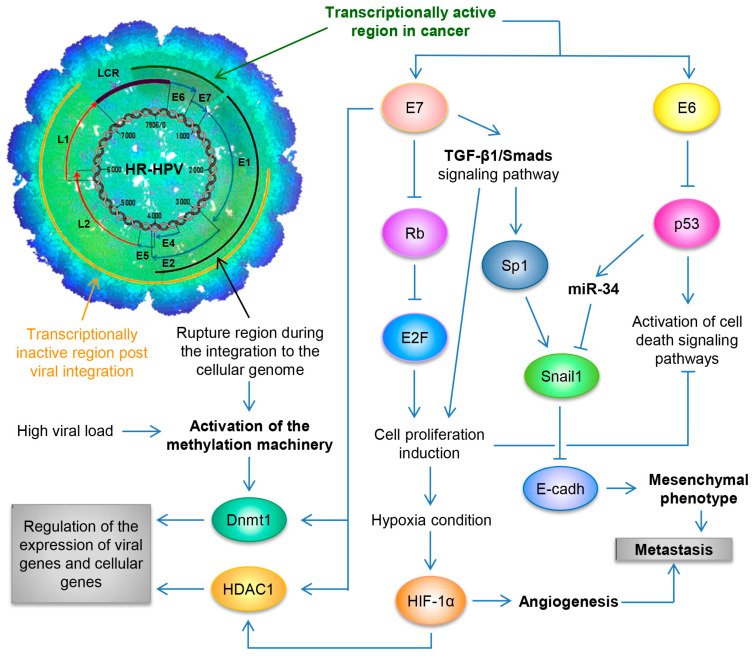
Schematic representation of the molecular mechanisms induced by HPV during the carcinogenic process of the uterine cervix. Both the viral load and the integration phenomenon induce the activation of the methylation machinery, which results in the regulation of the expression of viral genes and cellular genes. Loss of E2 function, either by methylation of the E2SB regions or by deletion of the viral genome during the integration phenomenon, causes the deregulated expression of the E6/E7 oncoproteins, which will consequently induce uncontrolled cell proliferation, evasion of cell death, activation of the angiogenic process and the acquisition of the mesenchymal or metastatic phenotype.

**Figure 3 biology-13-00077-f003:**
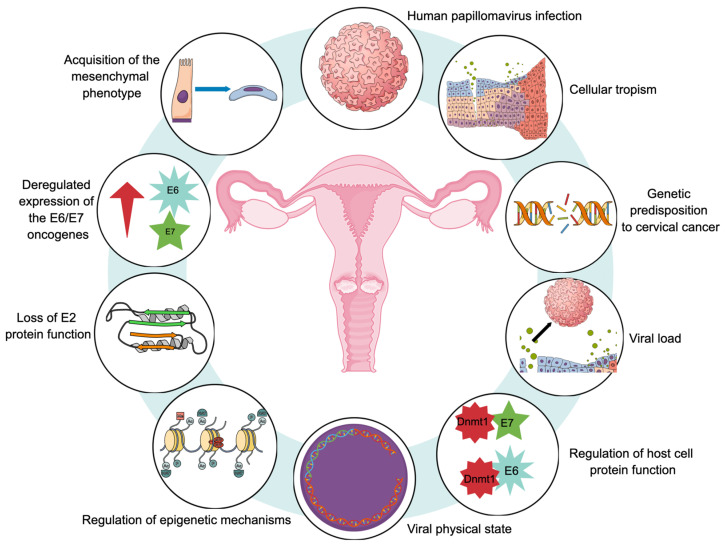
Proposed molecular hallmarks for cervical carcinogenesis. All of these markers would play an important role in the development of uterine cervical cancer.

**Table 1 biology-13-00077-t001:** Host cell proteins that interact with HPV proteins.

Modulated Mechanisms	Host Cell Proteins	HPV Protein	References
Increased cell proliferation	A4, Bap31, EGFR, ErbB4	E5	[126,127,128,129]
	CYLD, DLG1, DVL2, MPDZ, PTPN13, PTPN3	E6	[130,131,132,133,134,135,136,137,138,139]
	B-Myb/MuvB complex, BRG1, CDK2, CHD4, Cyclin A, Cyclin E, E2F1, E2F6, HDAC1, HDAC2, p107, p130, p27^KIP1^, pRb, PTPN14, SMAD1-4	E7	[140,141,142,143,144,145,146,147,148,149,150,151,152,153,154,155,156,157]
Evasion of the immune response	Calnexin, HLA-I heavy chain,	E5	[158,159]
	IRF3, TRIM25, TYK2, USP15, IRF3	E6	[160,161,162,163]
	IRF1, IRF9, IKKα, IKKβ, NLRX1, TAP1	E7	[164,165,166,167,168,169,170]
Loss of p53 function (due to inactivation or degradation)	hADA3, BCCIPβ, CBP, E6-AP, p300	E6	[106,107,118,119,121,171,172]
Loss of function of pRb (due to inactivation or degradation)	Calpain, Cullin 2, ZER1	E7	[173,174,175]
Inhibition of apoptosis	p53	E6	[118,176,177,178,179]
Defective DNA repair	BARD1, BRCA1, MGMT, XRCC1	E6	[180,181,182,183]
	BRCA1	E7	[182]
Epigenetic reprogramming	CARM1, PRMT1, SET7, DNMT1	E6	[184,185]
	p300, pCAF, SRC1, DNMT1, HDAC1, HAT	E7	[105,157,186,187,188,189]
Increased cell survival	AIF, BAK, Caspase-8, FADD, TNFR1	E6	[190,191,192,193,194]
	GSTP1, IGFBP-3, Siva	E7	[195,196,197]
Immortalization of host cell	c-Myc, hTERT, NFX1-123, NFX1-91	E6	[198,199,200,201]

Table adapted from Scarth et al. [202].

## Data Availability

Not applicable.

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
