# Peer review of "The Hallmarks of Cervical Cancer: Molecular Mechanisms Induced by Human Papillomavirus"

_biology, 2024, doi:10.3390/biology13020077_

Round 1

Reviewer 1 Report

Comments and Suggestions for Authors

The review article entitled “The hallmark of cervical cancer: molecular mechanisms induced by HPV” in which authors described the important possible molecular mechanisms of cervical cancer influenced by human papillomavirus. This review article is well structured and very well written. Figures are well described and in detail. However, there are few minor issues need to be addressed as provided below.

1. Line 117, add reference number 42 after the sentence ‘John Doorbar has extensively described the cellular tropism of HPV.’

2. Table 1, although authors provide in last line that the Table was adopted from Scarth et al., [113]. It will be better if authors provide a separate column with individual references for each mechanism. This will be easy for the readers to find in-depth details for the individual mechanism.

3. Better to add more information for one of the mechanisms “Viral load”. Provide some cohort data-based studies, check some related articles below.

i. Tao X., et al. Risk stratification for cervical neoplasia using extended high-risk hpv genotyping in women with asc-us cytology: a large retrospective study from ChinaCancer Cytopathol. (2022) 130:248–58. doi: 10.1002/cncy.22536.

 ii. Zhao X., et al. Role of human papillomavirus dna load in predicting the long-term risk of cervical cancer: a 15-year prospective cohort study in ChinaJ Infect Dis. (2019) 219:215–22. doi: 10.1093/infdis/jiy507.

 iii. Baumann A., et al. Hpv16 load is a potential biomarker to predict risk of high-grade cervical lesions in high-risk HPV-infected women: a large longitudinal french hospital-based cohort studyCancers. (2021) 13:4149. doi: 10.3390/cancers13164149.

4. imilarly add more information for the subtitle “Loss of E2 protein function”. Very limited information has been provided in this section.

Reviewer 2 Report

Comments and Suggestions for Authors

The manuscript is well written and organised. The authors have provided a  summary of ten mechanisms that are activated by HR-HPVs during cervical cancer development. Moreover, they have highlighted these ten mechanisms involved in at least seven of the fourteen cancer hallmarks. However, it would be beneficial if the authors could elaborate further how these mechanisms are linked to the hallmarks of cervical cancer, including the genome instability, sustained proliferative signalling, replicative immortality, resistance to cell death, and evasion of the immune response.

Additionally, it would be beneficial if the author included a discussion on the recent development or challenges of these HPV-activated mechanisms to be considered as molecular targets in the clinic.

Also, please include more up-to-date references. This would make the review more informative and aligned with the title.

 Minor

1)      Please include reference for “John Doorbar has extensively described the cellular tropism of HPV”. (Line 117)

2)      Please edit the format for Reference 14. (Line 100)

Round 2

Reviewer 2 Report

Comments and Suggestions for Authors

No further comments are needed for this manuscript.